# Fomite Transmission Follows Invasion Ecology Principles

Peihua Wang,[a] Xinzhao Tong,[b] Nan Zhang,[a,c] Te Miao,[a] Jack P. T. Chan,[a] Hong Huang,[d] Patrick K. H. Lee,[b] Yuguo Li[a,e]

[a]Department of Mechanical Engineering, University of Hong Kong, Hong Kong SAR, China
[b]School of Energy and Environment, City University of Hong Kong, Hong Kong SAR, China
[c]Key Laboratory of Green Built Environment and Energy Efficient Technology, Beijing University of Technology, Beijing, China
[d]Institute of Public Safety Research, Department of Engineering Physics, Tsinghua University, Beijing, China
[e]School of Public Health, University of Hong Kong, Hong Kong SAR, China

**ABSTRACT** The invasion ecology principles illustrated in many ecosystems have not yet been explored in the context of fomite transmission. We hypothesized that invaders in fomite transmission are trackable, are neutrally distributed between hands and environmental surfaces, and exhibit a proximity effect. To test this hypothesis, a surrogate invader, *Lactobacillus delbrueckii* subsp. *bulgaricus*, was spread by a root carrier in an office housing more than 20 participants undertaking normal activities, and the microbiotas on skin and environmental surfaces were analyzed before and after invasion. First, we found that the invader was trackable. Its identity and emission source could be determined using microbial-interaction networks, and the root carrier could be identified using a rank analysis. Without prior information, *L. bulgaricus* could be identified as the invader emitted from a source that exclusively contained the invader, and the probable root carrier could be located. In addition to the single-taxon invasion by *L. bulgaricus*, multiple-taxon invasion was observed, as genera from sputum/saliva exhibited co-occurrence relationships on skin and environmental surfaces. Second, the invader had a below-neutral distribution in a neutral community model, suggesting that hands accrued heavier invader contamination than environmental surfaces. Third, a proximity effect was observed on a surface touch network. Invader contamination on surfaces decreased with increasing geodesic distance from the hands of the carrier, indicating that the carrier's touching behaviors were the main driver of fomite transmission. Taken together, these results demonstrate the invasion ecology principles in fomite transmission and provide a general basis for the management of ecological fomite transmission.

**IMPORTANCE** Fomite transmission contributes to the spread of many infectious diseases. However, pathogens in fomite transmission typically are either investigated individually without considering the context of native microbiotas or investigated in a nondiscriminatory way from the dispersal of microbiotas. In this study, we adopted an invasion ecology framework in which we considered pathogens as invaders, the surface environment as an ecosystem, and human behaviors as the driver of microbial dispersal. With this approach, we assessed the ability of quantitative ecological theories to track and forecast pathogen movements in fomite transmission. By uncovering the relationships between the invader and native microbiotas and between human behaviors and invader/microbiota dispersal, we demonstrated that fomite transmission follows idiosyncratic invasion ecology principles. Our findings suggest that attempts to manage fomite transmission for public health purposes should focus on the microbial communities and anthropogenic factors involved, in addition to the pathogens.

**KEYWORDS** surface hygiene, invasive species, microbial interaction, built environment, disease transmission, surface ecosystem

Address correspondence to Yuguo Li, liyg@hku.hk, or Patrick K. H. Lee, patrick.kh.lee@cityu.edu.hk.

The authors declare no conflict of interest.

*[This article was published on 3 May 2022 with errors in the text and supplemental material. The errors were corrected in the revised version, posted on 5 May 2022.]*

Contaminated environmental surfaces are known to spread infectious diseases through contact and carriage of hands (i.e., fomite transmission). Studies investigating the strength and extent of fomite transmission have largely been conducted within the scopes of separate scientific disciplines. For example, epidemiologists focus on pathogen viability and spatial distribution (1–4), ecologists focus on the dispersal of microbiotas between individuals and environmental surfaces (5–10), and engineers focus on the mechanisms of transmission dynamics (11–15). Despite advances in our understanding of fomite transmission, pathogens are typically studied individually or in a nondifferentiated way from the dispersal of microbiotas.

There is increasing awareness of the value of applying invasion ecology principles (16–18) to understand the fate of pathogens following invasion into the ecosystems of humans (18–22), animals (23–25), and plants (26–29). As such, invasion ecology principles can serve as a foundation for developing improved pathogen management strategies. For example, the decreases in human gut microbial diversity after the use of antibiotics to treat *Clostridioides* (formerly *Clostridium*) *difficile* (19–21) and elimination of pathogens from water bodies via ozonation and UV irradiation (24, 25) can lead to an unexpected proliferation of other pathogens. These phenomena are in line with the diversity-invasibility hypothesis (16, 30), which posits that microbial diversity is inversely correlated with pathogen invasion success because of resource competition. Numerous pathogen management strategies have been developed to address this issue, such as introducing a nonpathogenic invader strain that can fill the niche vacancy (19), introducing ecological K-strategists (mature microbial communities) that act as keystone species to stabilize microbial interactions (23–25), and transplanting a healthy, homeostatic microbial community (20, 24). Despite the relevance of invasion ecology principles to the aforementioned ecosystems, similar principles have not yet been explored in the context of fomite transmission.

Adopting an invasion ecology framework that treats the surface environment as an ecosystem and pathogens as invaders may enhance our understanding of fomite transmission. Fomite transmission may follow idiosyncratic invasion ecology principles, an understanding of which could serve as a basis for the development of ecological approaches to manage fomite transmission. For example, if an invader exhibits co-occurrence relationships with other taxa, it is reasonable to conjecture that they share an emission source, e.g., sputum or feces, and wearing masks and improving restroom hygiene could manage their invasion. If an invader is neutrally distributed between hands and environmental surfaces, then interventions should focus on both hand and surface hygiene. If invader transmission is mainly driven by the touching behaviors of infected individuals, locating and quarantining these individuals should be prioritized.

We hypothesized that fomite transmission follows invasion ecology principles. That is, invaders in fomite transmission are trackable, are neutrally distributed between hands and environmental surfaces, and exhibit a proximity effect (Fig. 1). To test our hypothesis, we conducted two independent unsupervised field experiments in an office environment between 8 a.m. and 9 p.m. with 26 and 23 participants (Fig. 2a and b). The office environment was chosen because it offers a realistic simplification of a complex built environment, with manageable environmental variables, including temperature, humidity, hygiene, layout, and occupancy. *Lactobacillus delbrueckii* subsp. *bulgaricus* (*L. bulgaricus*), a nontoxic, non-human-origin, Gram-positive bacterium that shares cell envelope properties with common Gram-positive nosocomial bacterial pathogens (31) (e.g., methicillin-resistant *Staphylococcus aureus* and vancomycin-resistant *Enterococcus*), was selected as the surrogate invader (32). During each experiment, *L. bulgaricus* was introduced approximately every 30 min via the hands of an unrevealed root carrier. Microbiotas on skin and environmental surfaces were extensively sampled prior to invader introduction and at the end of the experiments. The touching behaviors of the participants throughout the experimental periods were recorded by cameras. We tested the trackability of the invader using a microbiota-invader network (MIN), a microbiota co-occurrence network (MCN), and a rank analysis, the neutrality of the invader during touches using a neutral community

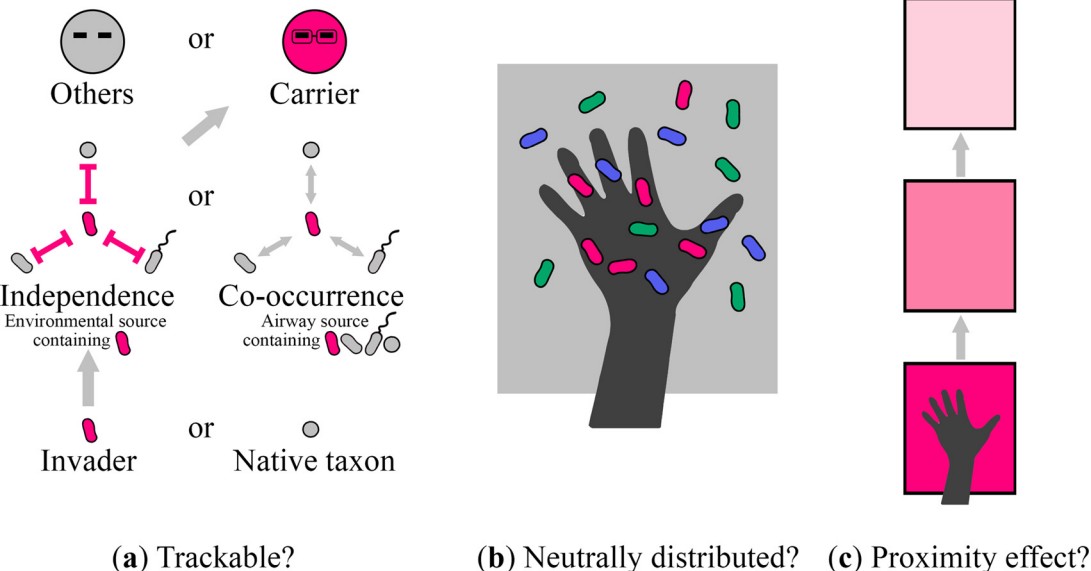

**(a)** Trackable?    **(b)** Neutrally distributed?    **(c)** Proximity effect?

**FIG 1** Hypotheses of invasion ecology principles in fomite transmission. Invaders in fomite transmission (a) are trackable (their identities and emission sources can be determined, and root carriers can be located), (b) are neutrally distributed between hands and environmental surfaces, and (c) exhibit a proximity effect.

model (NCM), and the invader proximity effect using a surface touch network (STN). Based on our findings, we argue that applying invasion ecology principles to fomite transmission may improve endemic forecasting, management strategy development, and the understanding of environmental public health.

## RESULTS

**Microbiota compositional drift through surface touches.** The offices were initially free of the invader. At the end of the experiments, hands, public surfaces, some objects on desks (e.g., cups), and chair surfaces (e.g., seatbacks) were highly contaminated with the invader, whereas faces and phones were the least contaminated surfaces (Fig. S1a to d). Figure 3a and b show the microbiota compositional drift on surfaces during the experiments. The surfaces on the principal coordinates analysis (PCoA) biplots are distributed in a ternary graph structure for both experiments, where the three points represent skin and inanimate surfaces highly contaminated with the invader (the arrow direction of *L. bulgaricus*), uncontaminated skin (bottom left), and uncontaminated inanimate surfaces (top left). Phones showed higher dissimilarities in microbiota composition than other inanimate surfaces and skin (Fig. S1e and f). During the 13-h experiment 1, microbiota homogenization was significant on hands (paired Mann-Whitney test, $P_{hand} = 2.2 \times 10^{-16}$) but not on faces or phones (Fig. 3c; many negative-control samples in experiment 2 were low in biomass and thus could not be sequenced to generate a meaningful density plot). The observed homogenization could be explained by the touching behaviors of the participants. Their hands were in contact with more surfaces (network degree in the STN), and with higher frequencies (network weighted degree in the STN), than their faces and phones (Table S1). We reported the detailed surface touch data in a previous publication (15).

**The invader was trackable in that its identity and emission source could be determined, and the root carrier could be located.** The results of MINs, MCNs, and the ranks of surface contamination with the invader by participant can be combined to reveal the identity of the invader, its emission source, and its root carrier in scenarios where no prior invader information is available. Probable invaders (i.e., those with the highest network degrees) were identified using the MINs (Fig. 4a and b). *L. bulgaricus* was identified as the most probable invader in both experiments (Table S2).

The MCNs revealed co-occurrence relationships during touches (Fig. 4c and d). The hub genera with the highest network degrees varied between the experiments. However,

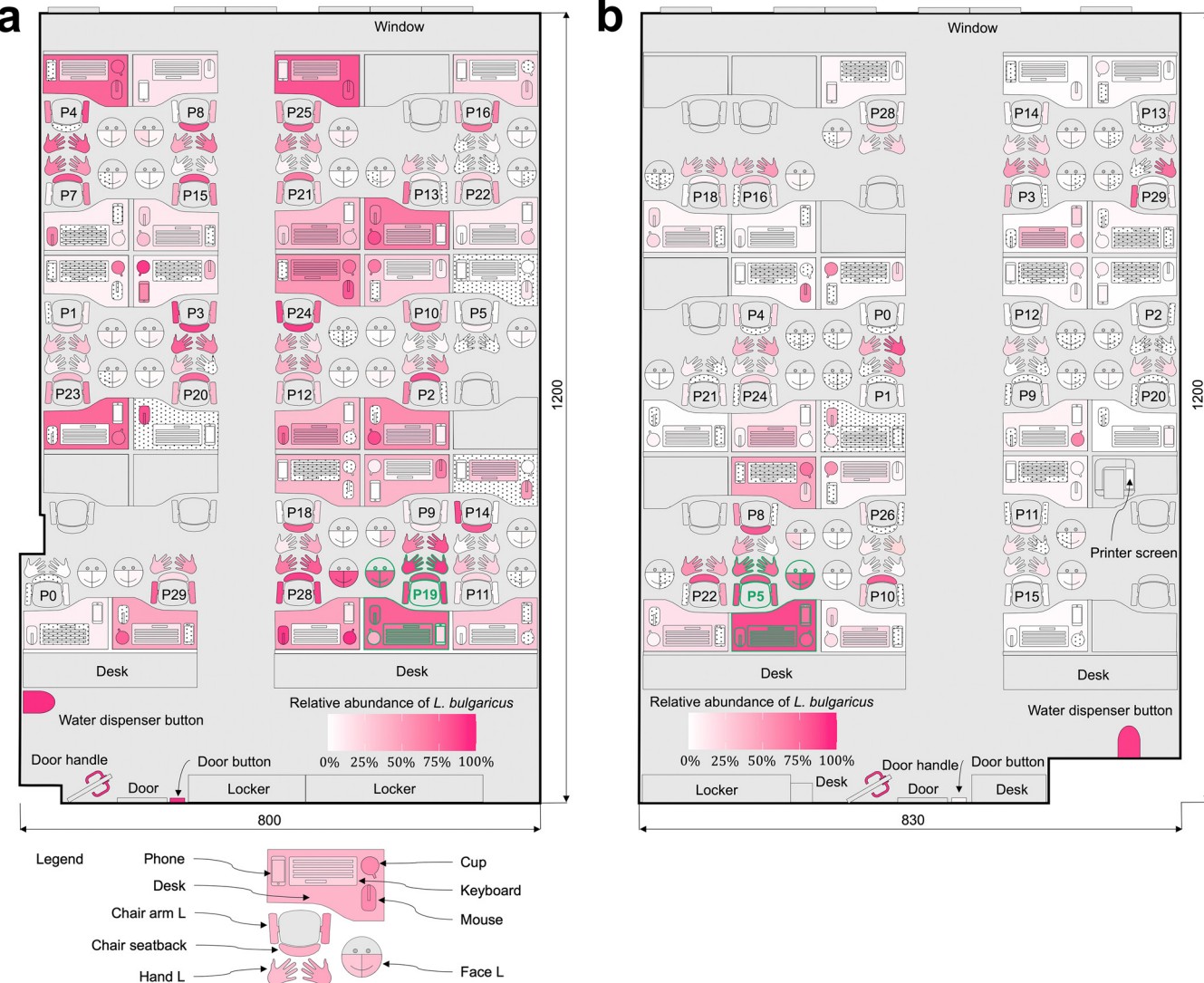

**FIG 2** Sampling locations and spatial distribution of the invader. Sampling locations and spatial distribution of the invader illustrated in schematic floor plans for (a) experiment 1 and (b) experiment 2. Surfaces shown in gray were not sampled. Surfaces with the dot pattern were sampled, but the corresponding samples had a low biomass and could not be sequenced. Surfaces outlined in green belong to carriers P19 and P5 in experiments 1 and 2, respectively. This schematic was adopted and modified from the floor plans previously reported by Wang et al. (15).

six genera (species level shown in Table S3)—*Streptococcus*, *Haemophilus*, *Neisseria*, *Actinomyces*, *Rothia*, and *Gemella*—were in the same network communities in both experiments (pairwise correlations shown in Fig. S2), indicating their co-occurrence relationships. As they were the most prevalent genera identified in sputum (33) and saliva (34), their co-occurrence relationships may originate from these common sources. The most probable invader, *L. bulgaricus*, was repulsed from the center of the force-directed MCNs due to its barely detectable co-occurrence relationship with other genera. This result indicates that, consistent with the experimental design, *L. bulgaricus* was emitted from an environmental source that exclusively contained *L. bulgaricus* rather than a human body source, which would result in a co-occurrence relationship with human native taxa, as demonstrated by the six sputum/saliva-associated genera.

Ranks of surface contamination with the highest relative abundance of *L. bulgaricus* by participant narrowed down the most probable root carriers (Fig. 4e). As expected, participants P19 (experiment 1) and P5 (experiment 2) were identified as the root carriers. P24 was also unexpectedly identified as a probable root carrier in experiment 1.

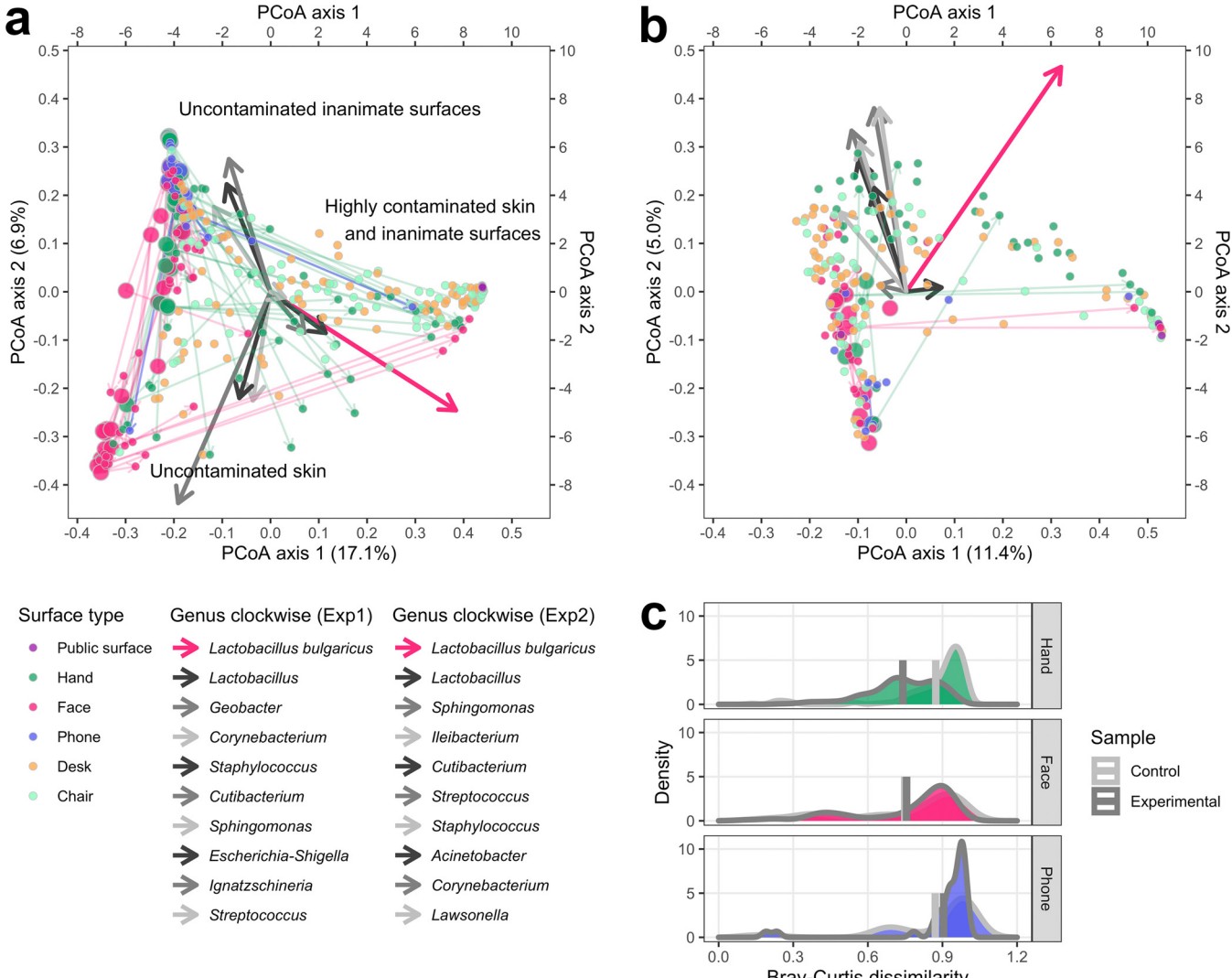

**FIG 3** Microbiota compositional drift through surface touches. Microbiota compositional similarities between surfaces indicated by Bray-Curtis dissimilarity and visualized by PCoA biplots for (a) experiment 1 and (b) experiment 2. Larger nodes represent negative-control samples (from hands, faces, and phones) collected before the experiments, and smaller nodes represent the samples collected at the end of the experiments. Arrows from larger to smaller nodes indicate the direction of microbiota compositional drift on the corresponding surfaces during the experiments. The surface type "desk" includes computer mice, keyboards, cups, and desks, and "chair" includes left and right chair arms and chair seatbacks. Arrows stemming from the coordinate origins denote the explanatory variables (the 10 most abundant genera in each experiment, including the invader *L. bulgaricus*) projected onto the PCoA coordination. (c) Density plot demonstrating microbiota homogenization on hands, faces, and phones during experiment 1. The vertical bars show the means. Invader ASVs were excluded from the analysis.

The STN bipartite projection showed that P24 had the highest interpersonal intimacy with P19, indicating that the two participants closely interacted during the experiment. These findings demonstrate that MINs, MCNs, and the ranks of invader contamination by participant can be used to determine the identity and emission source of the invader and the probable root carriers without prior invader information.

**The invader was below neutral in that hands accrued heavier invader contamination than environmental surfaces.** Using the NCM, we determined that the invader introduced in the experiments was in the below-neutral partition (75% of the invader amplicon sequence variants [ASVs] were below neutral for both experiments [Fig. 5a and b]; 74% and 75% were below neutral in subsample bootstrapping for experiments 1 and 2, respectively). That is, hands accrued heavier invader contamination than private inanimate surfaces, even with frequent handwashing (0.44 and 0.34 time/person/h for experiments 1 and 2, respectively). Similar to the statistically significant microbiota homogenization observed on hands (Fig. 3c), this phenomenon

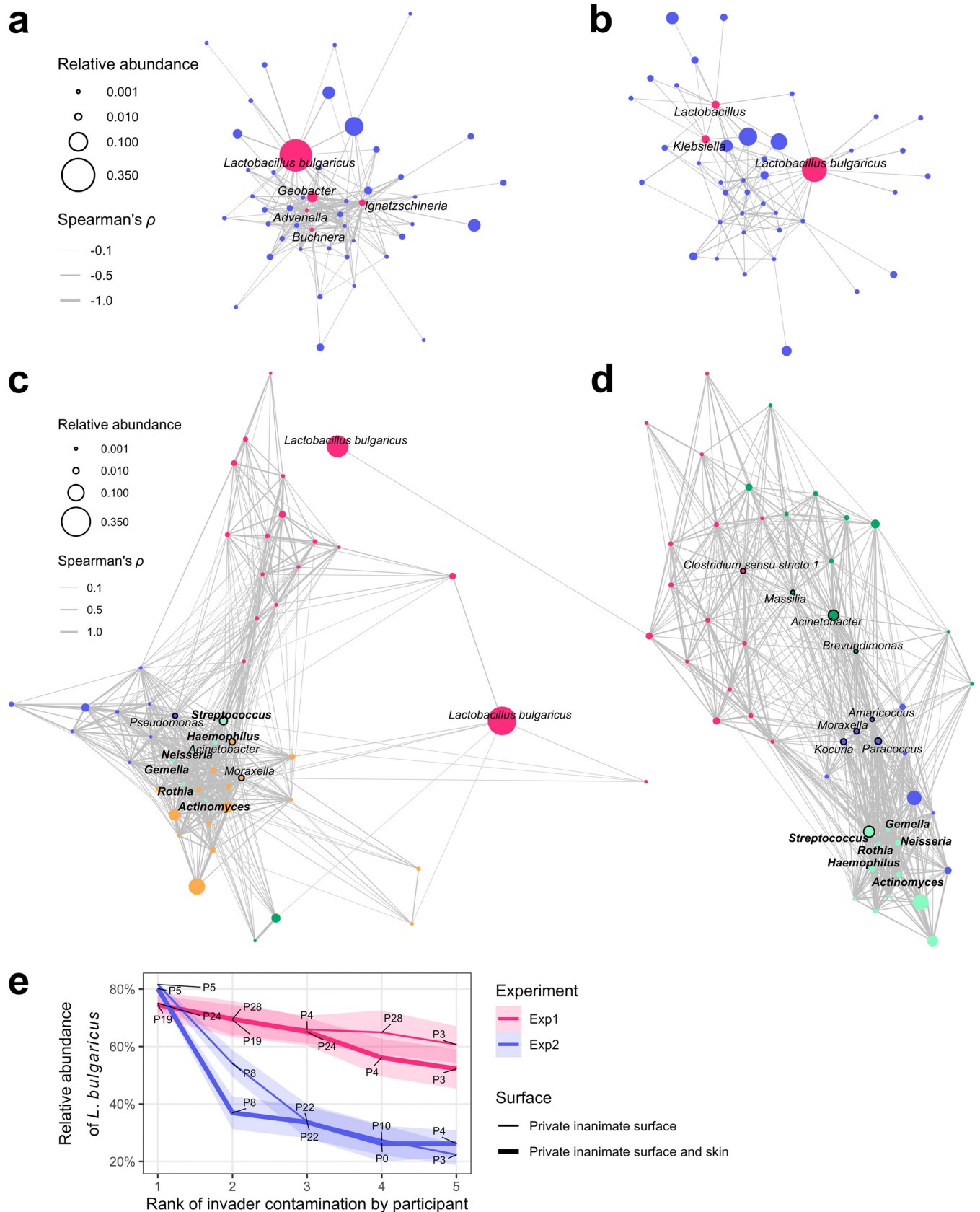

**FIG 4** The invader was trackable in that its identity and emission source could be determined, and the root carrier could be located. MINs with Fruchterman-Reingold force-directed layouts for the 50 most abundant genera (including the invader species *L. bulgaricus*) for (a) experiment 1 and (b) experiment 2. The

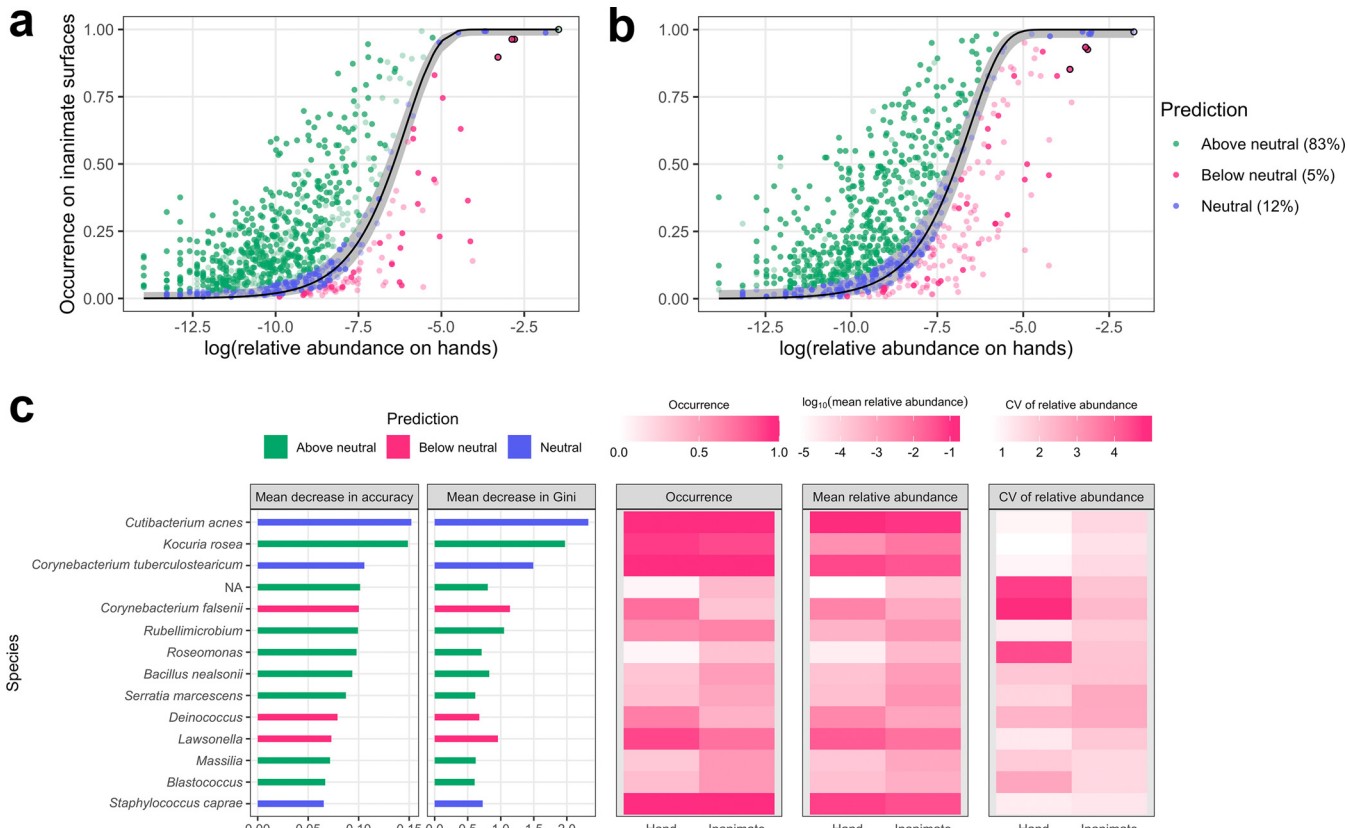

**FIG 5** The invader was below neutral in that hands accrued heavier invader contamination than environmental surfaces. Fit of the NCM at the ASV level using pooled hand samples as a source and private inanimate surfaces as the sink community for (a) experiment 1 and (b) experiment 2. Lines and shaded areas represent expected occurrences and 95% confidence intervals, respectively. The four invader ASVs are marked with black borders. The ASVs consistent in the NCM partitions between the experiments are highlighted. (c) Indicator ASVs that had statistical indicative powers for hands and private inanimate surfaces using a random forest model. The ASVs ranked in the top 20 for losses in both prediction accuracy and Gini index (feature variable impurity in the random forest model) when that particular ASV was removed from the random forest model are shown. CV, coefficient of variation.

could be explained by the touching behaviors of the participants. Their hands were in contact with more surfaces, and with higher frequencies, than private inanimate surfaces (detailed surface touch data were previously reported [15]). Sixty percent of the ASVs were consistent in NCM partitions between the experiments. Among these, 83% were above neutral.

Indicator ASVs, i.e., those that had statistical indicative powers for hands and private inanimate surfaces, were selected using a random forest model (Fig. 5c; error rate$_{inanimate}$ = 14%, error rate$_{hand}$ = 6%; area under curve = 0.98). The indicator ASVs for hands and private inanimate surfaces agreed with the NCM partition and appeared to be below neutral and above neutral, respectively. The exceptions were the ASVs assigned to *Cutibacterium acnes*, *Corynebacterium tuberculostearicum*, and *Staphylococcus caprae*, which were indicators for hands because of their higher occurrences (i.e., the proportion of samples in which an ASV was present) and mean relative abundances on hands, but they exhibited high dispersal abilities and were neutrally distributed between hands and private inanimate surfaces. Most indicator ASVs were above neutral (57%), similar to the NCM partition, in which most ASVs that were consistent in the NCM partitions between the experiments were

**FIG 4** Legend (Continued)
probable invaders with the highest network degrees are marked in red. Genera with no neighbors are not shown. MCNs with Fruchterman-Reingold force-directed layouts for the 50 most abundant genera (including the invader species *L. bulgaricus*) for (c) experiment 1 and (d) experiment 2. Names of the hub genera with the highest network degrees (vertices with black borders), six sputum/saliva-associated genera (bold genus names), and the invader are marked. Network communities are denoted by different vertex colors. (e) Invader contamination ranked by participant. Lines and shaded areas represent means ± 0.2 standard deviation. Participants P19 and P5 were the root carriers for experiments 1 and 2, respectively.

above neutral (83%), indicating a higher diversity of microbiotas on inanimate surfaces than on hands.

**The invader exhibited a proximity effect in that invader contamination on surfaces decreased with increasing geodesic distance from the hands of the carrier.** The ranks of surface contamination with the invader by participant show that the private inanimate surfaces of the root carriers accrued higher invader contamination than the surfaces of the other participants (Fig. 4e). This phenomenon was due to the invader proximity effect, in which invader contamination on private inanimate surfaces decreased with increasing geodesic distance (the lowest number of edges connecting two vertices) from the hands of the carrier in the STNs (Fig. 6a and b) (Dunn test: $P_{adj,Exp1,1-2} = 3.0 \times 10^{-7}$, $P_{adj,Exp1,2-3 \text{ or } +\infty} = 0.024$, $P_{adj,Exp2,1-2} = 0.044$, $P_{adj,Exp2,2-3 \text{ or } +\infty} = 0.672$ [Fig. 6c and d]). The proximity effect was also observed at the community level, as the microbiota contribution of carriers to private inanimate surfaces decreased with increasing geodesic distance from the hands of the carrier (Fig. 6e and f). The contributions of the surface owners' "best friends," who had the highest interpersonal intimacies with the surface owners in the STN bipartite projection, were consistent with the different geodesic distances (13% and 14% for experiments 1 and 2, respectively). The above-neutral microbiota made rare contributions, suggesting that although individual above-neutral ASVs were likely to be indicators of inanimate surfaces, they were collectively not compositionally similar to inanimate-surface microbiotas. High-touch private inanimate surfaces, as indicated by eigencentrality (a vertex measure that accounts for the centrality of a vertex and its neighbors), did not accrue higher levels of contamination than low-touch private inanimate surfaces (Fig. S3a to d).

## DISCUSSION

In this study, we applied invasion ecology principles to fomite transmission, demonstrating that the invader was trackable, was below neutral, and exhibited a proximity effect. Without prior invader information, the MINs, MCNs, and the ranks of surface contamination with the invader by participant jointly revealed that *L. bulgaricus* was the invader emitted from a source that contained no other bacteria and identified the probable root carriers who introduced the invader. The inference agreed with the experimental design that *L. bulgaricus* was the deliberately introduced invader. The NCMs demonstrated that hands accrued heavier invader contamination than environmental surfaces. Therefore, invader management requires hand hygiene routines. The STNs indicated that carriers' touching behaviors were the main drivers of invader fomite transmission. As carriers can be only retrospectively identified, the cleaning of public surfaces, including door handles and buttons, which accrued high levels of invader contamination, should be prioritized.

**Implementing targeted management strategies based on the co-occurrence relationships of an invader.** In addition to the single-taxon invasion demonstrated in the experiments, multiple-taxon invasion of an invader and human native taxa is also very likely due to phylogenetic (35) and physical mechanisms if an invader originates from inside human bodies. This phenomenon has been observed with *Streptococcus*, *Haemophilus*, *Neisseria*, *Actinomyces*, *Rothia*, and *Gemella*, as they are derived from common media (sputum [33] and saliva [34]) and demonstrate co-occurrence relationships. By extracting the co-occurrence relationships of an invader with other taxa in a microbial population using the MCN, we can substantially enhance our ability to identify the emission source of an invader and implement management strategies accordingly. For example, a recent study on the coronavirus disease 2019 (COVID-19) pandemic found that severe acute respiratory syndrome coronavirus 2 (SARS-CoV-2) had a co-occurrence relationship with *Rothia dentocariosa* in samples from hospitalized COVID-19 patients and environmental surfaces in the hospital environment, but the underlying mechanism was speculative (36). Using an MCN to overlay information about invaders with the underlying native microbiota composition, viewing SARS-CoV-2 as the invader and viewing *R. dentocariosa*, which demonstrates co-occurrence relationships with sputum/saliva microbiota, as a native species of the pulmonary tract and oral cavity, we suggest that the two species co-invaded the hospital environment in emitted sputum and saliva as a result of their co-occurrence relationship. In this case, wearing face masks, rather than hand and surface

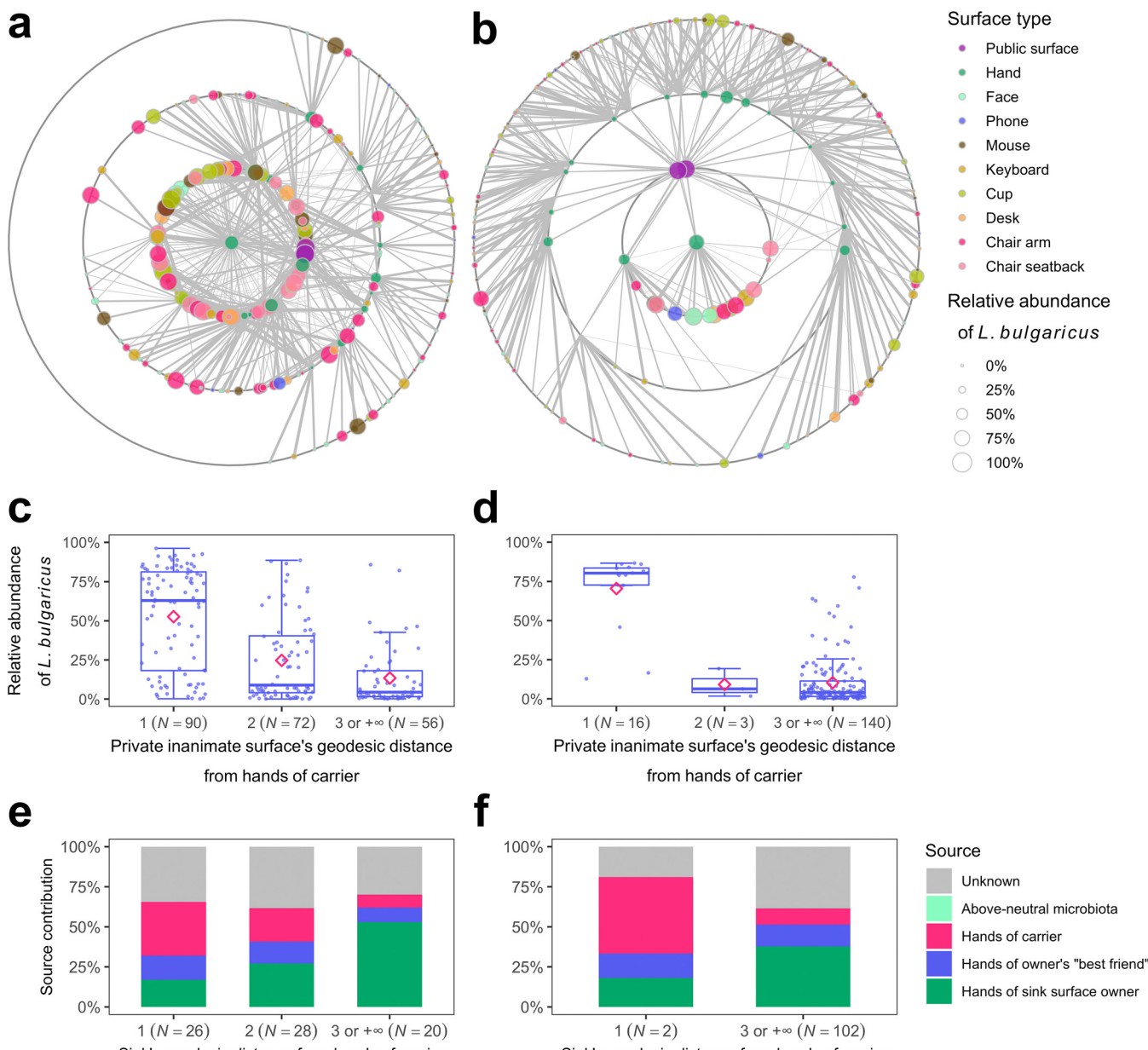

**FIG 6** The invader exhibited a proximity effect in that invader contamination on surfaces decreased with increasing geodesic distance from the hands of the carrier. STNs with focal layouts for (a) experiment 1 and (b) experiment 2. The vertices representing surfaces are arranged in concentric circles based on the geodesic distance from the hands of the carrier, which are in the center of the network. Surfaces with no neighbors are not shown. The edge width represents the $\log_{10}$-transformed number of touches between two surfaces. Box plots showing the distribution of the invader on private inanimate surfaces based on their geodesic distance from the hands of the carriers for (c) experiment 1 and (d) experiment 2. Box plots show the quartiles, and their means are represented by diamonds. Microbiota source tracking showing the contribution of sources to sinks (private inanimate surfaces) based on their geodesic distance from the hands of the carriers for (e) experiment 1 and (f) experiment 2. Note that the sample sizes for panels e and f are smaller than those for panels c and d, respectively, because only the sinks in which the sources, i.e., the owner, the owner's "best friend," and the carrier, were three different individuals were analyzed to avoid duplication.

hygiene measures, should be prioritized to manage the SARS-CoV-2 invasion. Similarly, in cases where co-occurrence relationships are identified between an invader and feces-associated taxa in environments with poor hygiene, restroom hygiene should be prioritized.

**Improving the understanding of environmental public health by adopting an invasion ecology framework.** NCMs can serve as theoretical frameworks for monitoring the invasion ecology of fomite transmission over time and identifying the collapse tipping point of a surface ecosystem at which an invader becomes established on the environmental

surfaces. When an invader is first introduced, it appears to be below neutral. As active invasion proceeds, the mean relative abundance of the invader on environmental surfaces and hands may gradually decrease because environmental factors such as temperature, humidity, and resource availability could become unfavorable to the growth of the invader in new environments (30). However, the occurrence of the invader on environmental surfaces may remain stable if the invader evolves phasic lifestyles to prevent die-off (30). These processes are reflected in the NCM as a switch in invader neutrality from below neutral (selected by hands) to above neutral (selected by environmental surfaces). Therefore, the switching of invader neutrality serves as the collapse tipping point, reflecting active invasion on environmental surfaces to establish an initial invader population.

The native microbiotas on environmental surfaces, which had a higher diversity than the microbiotas on hands in our experiments, tend to exhibit resilience against fomite transmission in the long term. This is supported by the diversity–invasibility hypothesis, because microbial communities with higher diversity are more likely to host species that are superior to invaders in terms of competition for resources (17). In particular, indicator species, as the ecological K-strategists that facilitate interspecies interactions, may have an important role in shaping resilience (25). The diversity–invasibility relationship also informs the feasibility of managing ecological fomite transmission. Although the traditional approaches of broad-spectrum disinfection and frequent cleaning have greatly improved surface hygiene, they also alter surface ecosystems by decreasing microbial diversity, leaving surfaces vulnerable to invasion. Instead, ecological approaches that build the resilience of a surface ecosystem against invaders should focus on surface materials (for microbial adhesion and biofilm formation) (37) and mutualism within the surface microbial community. Ecological approaches to improve mutualism include introducing a nonpathogenic invader strain to fill the niche vacancy (19, 38), introducing a group of ecological K-strategists (23–25), and transplanting a healthy microbial community to ensure resource utility (20, 24). However, caution should be exercised to avoid the opportunistic initiation of infection during introduction or transplantation processes and the unexpected emergence of antimicrobial-resistant microorganisms through horizontal gene transfer (39).

The invader proximity effect highlights anthropogenic "natural" dispersal (i.e., carriers' touching behaviors) in invasion ecology during fomite transmission. Therefore, the invasion ecology of fomite transmission differs from that observed in ecosystems where an invader is likely to be introduced by natural dispersal. An active carrier may support the continuous dispersal of an invader (termed the "introduction effect" or "propagule pressure" [40]) for its initial establishment on environmental surfaces against stochastic events (17). In other words, a human-mediated introduction effect may be a frequent feature of fomite transmission. As a result, the invasion success of fomite transmission, reflected by invader population growth and site expansion, may exceed 0.1% (baseline as estimated by the tens rule [16, 28], an invasion ecology hypothesis) because a high introduction effect may temporarily uncouple the diversity-invasibility relationship for invasion (16, 40).

**Implications for environmental public health.** In summary, we demonstrated the potential of adopting an invasion ecology framework that synthesizes epidemiological and ecological efforts to enhance our ability to manage fomite transmission in our simulated experiment. We also demonstrated an application of the invasion ecology framework in the real-world scenario by analyzing and incorporating the results of a case from the COVID-19 pandemic (36). We found that the identity and emission source of the invader could be determined, and the probable root carriers could be identified. Different targeted environmental intervention strategies should be implemented based on invader co-occurrence relationships (e.g., maintaining hand and surface hygiene or wearing masks). We also found that the invader was below neutral in the NCM, with hands accruing heavier invader contamination than environmental surfaces. The microbiotas on environmental surfaces, which has a higher diversity than the microbiotas on hands, tend to shape resilience against fomite transmission in the

long term. The NCM serves as a theoretical framework to capture the tipping point of a surface ecosystem at which an invader becomes established on environmental surfaces. Within this framework, the switching of invader neutrality from below neutral to above neutral represents the tipping point. Moreover, the invader exhibited a proximity effect. The carriers' touching behaviors were the main driver of fomite transmission. The human-mediated introduction effect distinguishes the invasion ecology of fomite transmission from that of other ecosystems. Although we demonstrated that fomite transmission follows invasion ecology principles, we conducted only two 13-h experiments with extensive sampling in an office environment. Long-term experiments in health care and household environments that track an invader from dispersal to final establishment or die-off are warranted to further elucidate the invasion ecology of fomite transmission. A more exclusive surrogate invader that is not associated with built environments should be used to reduce the potential risks of background contamination. Moreover, more advanced tools, such as metagenomic and metatranscriptomic sequencing, should be applied to assess viable microbial populations, opportunistic invaders, and cross-domain microbial interactions at the strain level. Nevertheless, this study provides a foundation for developing ecological approaches to manage fomite transmission.

## MATERIALS AND METHODS

**Experimental design.** This study was approved by the Human Research Ethics Committee of the University of Hong Kong (reference number EA1811019). Informed consent was obtained from all participants.

To test our hypothesis, the externally introduced bacterium *L. bulgaricus* (ATCC 11842) was selected as a surrogate invader for common Gram-positive bacterial pathogens. Two independent unsupervised field experiments were conducted in two student offices on two consecutive sunny Saturdays between 8 a.m. and 9 p.m. at Tsinghua University, Beijing, China. The first experiment had 26 participants, and the second had 23. All participants were graduate students or visiting scholars from the university who were informed of the purpose of the study but blind to the invader source location. Surface touch data and surface microbiota data were collected. To collect the touch data, the participants' activities were recorded using 22 ceiling cameras and analyzed at a 1-s resolution (41). Detailed information about the touching behaviors of the participants was previously reported (15). To collect the microbiota data, each test surface was initially sampled using two swabs (552C; Copan, USA): a wet swab moistened with phosphate-buffered saline (806544; Sigma-Aldrich, USA) and a dry swab to collect the residual liquid. The sampled surface areas are listed in Table S4. Genomic DNA was extracted using a DNeasy PowerSoil kit (Qiagen, Germany), and the V3-V4 regions of the 16S rRNA genes were amplified using the primers 341F and 806R and analyzed by amplicon sequencing (Novogene, China). The resulting raw amplicon sequencing data were used to infer ASVs using the R package dada2 (v1.16.0) (42) (details provided in Text S1). Microbiota data quality control was performed as described in Text S1. After the ASV inference and microbiota data quality control steps, an average of 64,181 high-quality sequence reads per sample were retained (62.8% of raw reads). Microbiota data diagnostics included rarefaction curves (Fig. S4a), which demonstrated that the alpha diversities converged at a rarefaction depth of 30,000 reads, and sequencing depth distribution (Fig. S4b), which demonstrated that most samples met the sequencing depth threshold of 30,000 reads. The invader *L. bulgaricus* was identified based on four ASVs (99.5% similar) detected in the stock solutions.

The night before the experiments were performed, the surfaces in the offices were thoroughly cleaned with a 2% sodium hypochlorite solution (239305; Sigma-Aldrich, USA). Before beginning the experiments, the background invader contamination levels in the offices were measured by randomly collecting samples from five desks, five computer mice, two water dispenser buttons, two door handles, one printer screen, and one printer button (printer samples were collected only in experiment 2) for each experiment. Samples were analyzed via a quantitative PCR assay with a detection limit of ~1,000 copies of the *L. bulgaricus* proline iminopeptidase (*pepIP*) gene per sample (32). The copy numbers of the *pepIP* gene in these samples were all below the detection limit. The *L. bulgaricus* genome has one copy of *pepIP* gene, hence the *pepIP* gene copy number is equivalent to the number of *L. bulgaricus* cells. The participants consented to refrain from consuming certain dairy products the day before and during the experiments to avoid contaminating the test sites with food-derived *L. bulgaricus*.

During the experiments, one participant was selected as the root carrier to spread the invader by applying a 1.5 mL stock solution (preparation method described by Wang et al. [32]) containing approximately $1.2 \times 10^7$ cells of *L. bulgaricus* to the carrier's hands approximately every 30 min. Before entering the offices, the participants cleaned their hands, faces, and phones with disinfectant wet wipes, and three samples (one for both hands, one for both sides of the face, and one for the phone) were collected as negative controls. Distilled water was provided for the participants to rinse their mouths before entering the offices for the first time and after meals. Surface samples (from public inanimate surfaces, including door handles, door buttons, water dispenser buttons, and a printer screen [only in experiment 2]; private inanimate surfaces, including phones, computer mice, keyboards, cups, desks, chair arms, and chair seatbacks; and skin on the hands and face) were collected at the end of the experiments. For participants who left early, samples were collected from their hands, faces, and phones immediately before they left. Samples from unoccupied desks were used as negative controls to identify environmental contaminant ASVs.

 10.1128/msystems.00211-22 **11**

**Microbiota invader network and microbiota co-occurrence network.** Pairwise similarity-based MINs and MCNs (35, 43) were applied to identify invaders that were normally absent in the test environment and co-occurrence relationships between genera, respectively. The networks were constructed using the 50 most abundant genera (including the invader species) as vertices and Spearman's $\rho$ values between genera as edges. First, a genus table was constructed from an ASV table, in which ASVs unassigned at the genus level were removed and ASVs assigned to the same genus were pooled. Then, $\rho$ values between genera were calculated as cor.test(genus$_i$, genus$_j$, method = "spearman"), excluding surfaces on which both genus$_i$ and genus$_j$ were zero, which were removed beforehand. Statistical significance levels from multiple tests were adjusted using the false discovery rate, and $\rho$ values with adjusted significance levels equal to or greater than 0.05 were removed from the downstream MIN and MCN construction. The network characteristics were calculated using the R package igraph (v1.2.6) (44).

The MINs were constructed using negative $\rho$ values as edges. In the MINs, each genus was treated as the invader once, and the other genera were treated as native surface genera. We assumed that the mean relative abundance of a native genus was $p_{native}$, with a small variance before the invader was introduced to the test environment. Once an invader was introduced, $p_{native} = q_{native}/(1 - q_{invader})$, where $q$ is the relative abundance of a genus on an invader-contaminated surface. That is, $q_{native}$ and $q_{invader}$ were negatively correlated because $q_{native} = -p_{native} \times q_{invader} + p_{native}$. The negative correlation was indicated by a negative $\rho$ value. An invader was expected to have the highest number of negative correlations with the other genera, i.e., the highest network degree in an MIN.

The MCNs were constructed using positive $\rho$ values as edges. In the MCNs, we assumed that the absolute abundances of two genera with a co-occurrence relationship were positively correlated, and their correlation could be measured by positive $\rho$ values. Using the relative abundances of genera obtained from amplicon sequencing, the same $\rho$ value was obtained as when absolute abundances were used because $\rho$ measures the rank correlation. A group of genera with co-occurrence relationships was expected to demonstrate pairwise positive correlations and, therefore, form a network community in an MCN. Greedy optimization of network modularity (a structural measure that indicates the strength of a division of a network into communities) was used for community detection. In practice, $\rho^3$ values were used as edges to increase edge weight heterogeneity, providing a finer division of the network into communities. The genera that displayed the strongest co-occurrence relationships, i.e., were in the same community for each experiment, and the communities that demonstrated the greatest similarity between the experiments, as measured by length(intersect(community$_i$, community$_j$))/length (union(community$_i$, community$_j$)), were selected.

**Neutral community model.** An NCM (45–47) was built to determine the neutrality of each ASV during touches. The NCM hypothesizes that dispersal and demographic drift (stochastic deaths and births), instead of environmental selection, have a key role in shaping the structure of a microbial community. Briefly, an ASV with a higher relative abundance in a source was expected to have a higher occurrence in a sink community if dispersal and demographic drift were important ecological forces. In this study, hand samples were pooled as a single source, and private inanimate surfaces constituted the sink community (excluding source and sink samples belonging to the carrier). To determine the expected occurrence of each ASV (within the 854 most abundant ASVs) in the sink community, first, the relative abundance of each ASV in the pooled source was noted as the probability $P_{ASV}$. The cumulative beta distribution with the upper tail was then used to formulate the expected occurrence for each ASV in the sink community as $p$beta(threshold of relative abundance = $1/N$, shape$_1 = N \times P_{ASV} \times m$, shape$_2 = N \times (1 - P_{ASV}) \times m$), where $N$ is the number of ASVs in a sample and a single free parameter subject to optimization, $m$, describes the probability of dispersal from a source following a stochastic death within a sink, as opposed to $1 - m$, the probability of a stochastic birth.

Next, maximum-likelihood estimation was used to optimize $m$ and determine the expected occurrence for each ASV. The detailed formula derivation and numerical implementation method were previously described by Sloan et al. (46) and Burns et al. (47), respectively. Finally, the variabilities of the expected occurrences were calculated using 95% binomial Wilson score confidence intervals. The above-neutral ASVs (above the upper bound of the confidence intervals), and the below-neutral ASVs (below the lower bound of the confidence intervals) were considered overrepresented and underrepresented, respectively, in the sink community. That is, the NCM hypothesis was rejected, and these ASVs were shaped mainly by ecological selection by the sink and source environments, respectively. Likewise, the ASVs that fell within the confidence intervals were neutrally distributed. That is, the NCM hypothesis could not be rejected, and these ASVs were shaped mainly by dispersal and demographic drift. The neutralities of the invader ASVs were further verified by bootstrapping at a 90% subsample size with 1,000 simulations.

**Surface touch network.** STNs were constructed to investigate the invader proximity effect (15). A vertex represented a surface (two hands of a participant were pooled as one surface), and an edge represented the $\log_{10}$-transformed number of touches between two surfaces. The geodesic distance from the hands of the carrier and eigencentrality for each surface were calculated, and their relationships with invader contamination were investigated. Invader contamination was measured by the relative abundance of the invader at the species level and the microbiota contribution from the carrier at the community level using microbiota source tracking.

Network bipartite projection onto hands was used to identify the "best friends" of the participants for microbiota source tracking. The projection function measured interpersonal intimacies as follows:

$$W_{u,v} = \sum_{nbr} (W_{u,nbr,direct} \times W_{v,nbr,direct})^{\frac{1}{4}} + W_{u,v,direct}$$

where $u$ and $v$ are two hands representing two participants, nbr is a neighbor surface of both $u$ and $v$, $W_{direct}$ is the edge weight between two surfaces, and $W$ is the projected edge weight between two

hands. A higher value for $W$ indicated a higher level of intimacy. The best friend of participant $u$ was defined as participant $v$, who had the highest $W_{u,v}$ for $u$.

**Other models and statistical tests.** PCoA biplots were constructed using the R packages vegan (v2.5-6) (48) and ape (v5.4-1) (49) to determine the microbiota compositional drift through surface touches (details provided in Text S1). A random forest model was built using the R package randomForest (v4.6-14) (50) to identify the indicator ASVs that had statistical indicative powers for hands and private inanimate surfaces (details provided in Text S1). Microbiota source tracking was conducted using the R package FEAST (v0.1.0) (51) to investigate the invader proximity effect at the community level (details provided in Text S1). All statistical tests were conducted in R (v3.6.2) (52). Data visualization was conducted using the R package ggplot2 (v3.3.2) (53).

**Data availability.** Raw sequence reads (550 samples) from this study have been deposited in the NCBI SRA under BioProject no. PRJNA725971.

## SUPPLEMENTAL MATERIAL

Supplemental material is available online only.

**TEXT S1**, DOCX file, 0.03 MB.
**FIG S1**, TIF file, 2.8 MB.
**FIG S2**, TIF file, 0.4 MB.
**FIG S3**, TIF file, 2.4 MB.
**FIG S4**, TIF file, 1.8 MB.
**TABLE S1**, XLSX file, 0.01 MB.
**TABLE S2**, XLSX file, 0.01 MB.
**TABLE S3**, XLSX file, 0.01 MB.
**TABLE S4**, XLSX file, 0.01 MB.

## ACKNOWLEDGMENTS

This study was supported by the HK Research Grants Council Collaborative Research Fund (grant number C7025-16G) and the University of Hong Kong-Zhejiang Institute of Research and Innovation Seed Funding Scheme (grant number 04004).

We thank Pengcheng Zhao at the University of Hong Kong for the constructive criticism of the manuscript.

We declare no competing interests.

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
