## [Reviewer comments · mSystems]

Fomite transmission follows invasion ecology principles

Peihua Wang, Xinzhao Tong, Nan Zhang, Te Miao, Jack Chan, Hong Huang, Patrick Lee, and Yuguo Li

Corresponding Author(s): Yuguo Li, The University of Hong Kong

Review Timeline:

Submission Date:	March 4, 2022
Editorial Decision:	March 29, 2022
Revision Received:	April 13, 2022
Accepted:	April 14, 2022

Editor: Suzanne Ishaq

Reviewer(s): Disclosure of reviewer identity is with reference to reviewer comments included in decision letter(s). The following individuals involved in review of your submission have agreed to reveal their identity: Ryan Andrew Blaustein (Reviewer #3)

Transaction Report:

DOI: <https://doi.org/10.1128/msystems.00211-22>

March 29, 2022

Prof. Yuguo Li
The University of Hong Kong
Mechanical Engineering/Public Health
Pokfulam Road
Hong Kong
China

Re: mSystems00211-22 (Fomite transmission follows invasion ecology principles)

Dear Prof. Yuguo Li:

Thank you for submitting your manuscript to mSystems. We have completed our review and I am pleased to inform you that, in principle, we expect to accept it for publication in mSystems. However, acceptance will not be final until you have adequately addressed the reviewer comments.

The authors put a great deal of effort into revising their original manuscript, and the previous reviewer and I appreciated the consideration of the comments from review 1. One of the previous reviewers was satisfied by the revisions, but the other could not be reached so I solicited an additional review. Reviewer 3 provided feedback about the framing of the paper, the presentation of the figures, and several other details, and I think these would further strengthen the paper.

Preparing Revision Guidelines

Sincerely,

Suzanne Ishaq

Editor, mSystems

Journals Department
Reviewer comments:

Reviewer #2 (Comments for the Author):

Thank you for a thorough response to my comments. Very interesting work!

Reviewer #3 (Comments for the Author):

see attached

Wang et al. present an interesting experiment on the transmission of an introduced bacterial contaminant through an office environment with context to the native host and surface microbiota. The main results are in line with expectations; e.g., (i) participant hands became the most frequently “contaminated” site sampled (probably because they have the highest degree of touch network relative to other sites sampled) and (ii) proximity effect of source of transmission. While the experimental design is sound, and this could be a nice benchmarking study for site-specific bacterial transmission, the broader invasion ecology applications and explanations for findings seem speculative. Some re-direction may help improve the manuscript prior to publication. Comments below.

1. The implication of invasion ecology driving the observed microbiota transitions seems like a reach for the scope of the experiment. Continuous introduction of the contaminant bacterial strain every 30 mins, along with the network of surface touching (stemming outward from the carrier) are probably the primary drivers of contaminant transmission. As such, the flux in relative abundances of host/surface microbiota observed between the two time points (morning and end of day) are probably just related to the physical introduction of *L. bulgaricus*, mainly at high-touch surfaces. I am not sure it is “invading” as much as it is physically tracking across space and time. There is discussion about ecology and evolution in the text, though *L. bulgaricus* is most likely passively dispersed and not growing in the environments being sampled (or at least was not measured to be). Do the data show otherwise?
2. I think the focus of the paper would be stronger if re-framed as a benchmarking experiment to model fomite transmission using 16S amplicon abundances to get at questions like: How accurately can we model transmission with these data, and what are parameters for proximity effects in this specific space? What is the variation across the experimental systems? How does this relate to other fomite transmission models (I am not an epidemiologist by the way, so am personally not sure)?
3. The authors’ previous report is similar: <https://www.sciencedirect.com/science/article/pii/S0304389421011018#fig0005>. Fig. S1 looks nearly identical to the Fig. 1 from that report, with the key difference being quantification method of *L. bulgaricus* (i.e., amplicon relative abundance here vs. single-gene qPCR copy number there). Please expand on key differences to validate novelty in publishing the current study.
4. The results and figures present contaminant transmission as measured relative abundances among microbiota, sometimes among relative abundances of genera, species, and ASVs. Why are different taxonomic levels used throughout the paper as means for the quantification? Resolution should probably not go finer than species, considering multiple ASVs map to *L. bulgaricus* (which may even suggest potential problems for tracking it as a model strain). Please be consistent or explain.
5. The introduction and discussion provide considerable focus on phenomena that may lack relevance to the specific context of the study. Pulmonary/sputum/oral/gut microbiomes are mentioned, even as sources to the observed environmental sample, but they were not sampled here. To my knowledge, built environment surface microbiomes look mostly like a mix of environmental and skin-derived microbiota, and that is probably what is seen here as well.
6. Please expand on limitations in the discussion.
7. Thank you for making the sequencing data public.

Specific comments:

- L. 43: Remove ”;” and just split the sentences.
 - L. 52: What do you mean by “underlying dynamics”? Maybe replace with “context”
 - L. 90-92: While this statement makes sense, I am not sure bacteria are really growing on indoor surfaces (at least not much), like they do in the gut. Modeling pathogen transmission and persistence seems like a more appropriate focus to your dataset/design.
 - L. 122-124: This may be a stretch. How do you suggest using this framework?
 - L. 161-164: I thought it was emitted from root carrier’s hands. Was that the initial source? Also, this suggestion becomes somewhat speculative, since different participants may harbor variation in their respective skin microbiota.
 - L. 165-168: How do you have two probable root carriers in Exp 1 (a single participant must have the highest rank in abundance)? Was one presumed by measured abundance on hands, and another by abundance on something else?
 - L. 208: “best friends”?
 - L. 223: What is meant by emitted from a source that contained no other bacteria? The environment and the participant skin (root carrier hands) are not sterile
 - L. 236-237: These taxa are simply human-associated. They occur in the oral microbiome, but also on skin. Transmission from touching is probably the most likely explanation in the experiment. Sputum explanation seems irrelevant (was sputum sampled?).
 - L. 241-254: How does hygiene fit into your experiment? Not sure hand washing and mask wearing were controlled? Is this relevant?
 - L. 262-263: The more basic explanation may be that invading microbes likely do not “grow” in the environment.
 - L. 269-273: A simpler explanation may just be how this is related to hands being the primary vector for transmission (i.e., a point of contact to all surfaces in the study), and *L. bulgaricus* being repeatedly introduced on hands. Environmental surfaces should look less changed if they are not as frequently contacted by the hands of any initial carrier or subsequent carrier.
 - L. 303: What data analysis gets to this conclusion? Don’t see anything in the results
-
- Fig. 1: Good idea to illustrate principles. I suggest complementing this figure with a schematic for sampling/experimental design to show how you tested A/B/C.
 - Fig. S1: Seems more important to the story. Perhaps add to Fig. 1.
 - Fig. 2: Gray arrows in the genus list are confusing. Maybe need a different color scheme if the idea is to portray the specific taxa associations in the PCoA.
 - Fig. 3: Needs clarification. What do the colors indicate -- same indication in ABCD? What about E? Also, what do you mean by private vs. private inanimate?
 - Fig. 4: the CI’s seem very tight for the data points displayed. How was this modeled? What is the color transparency in A and B indicating? Plotting species in A and B should be done (rather than ASVs) to avoid confusion and to be consistent with C.

April 14, 2022

Prof. Yuguo Li
The University of Hong Kong
Mechanical Engineering/Public Health
Pokfulam Road
Hong Kong
China

Re: mSystems00211-22R1 (Fomite transmission follows invasion ecology principles)

Dear Prof. Yuguo Li:

Your manuscript has been accepted, and I am forwarding it to the ASM Journals Department for publication. For your reference, ASM Journals' address is given below. Before it can be scheduled for publication, your manuscript will be checked by the mSystems production staff to make sure that all elements meet the technical requirements for publication. They will contact you if anything needs to be revised before copyediting and production can begin. Otherwise, you will be notified when your proofs are ready to be viewed.

Publication Fees:

We recognize that the video files can become quite large, and so to avoid quality loss ASM suggests sending the video file via <https://www.wetransfer.com/>. When you have a final version of the video and the still ready to share, please send it to mSystems staff at mSystems@asmusa.org.

For mSystems research articles, if you would like to submit an image for consideration as the Featured Image for an issue, please contact mSystems staff at mSystems@asmusa.org.

Sincerely,

Suzanne Ishaq
Editor, mSystems

Journals Department
Table S1: Accept
Table S4: Accept
Fig. S4: Accept
Fig. S1: Accept
Fig. S2: Accept
Table S2: Accept
Text S1: Accept
Table S3: Accept
Fig. S3: Accept